# Pulmonary Tuberculosis and Risk of Lung Cancer: A Systematic Review and Meta-Analysis

**DOI:** 10.3390/jcm11030765

**Published:** 2022-01-30

**Authors:** Soo Young Hwang, Jong Yeob Kim, Hye Sun Lee, Sujee Lee, Dayeong Kim, Subin Kim, Jong Hoon Hyun, Jae Il Shin, Kyoung Hwa Lee, Sang Hoon Han, Young Goo Song

**Affiliations:** 1Yonsei University College of Medicine, Seoul 06273, Korea; sooyoungsarah@gmail.com (S.Y.H.); crossing96@naver.com (J.Y.K.); 2Biostatistics Collaboration Unit, Yonsei University College of Medicine, Seoul 06273, Korea; hslee1@yuhs.ac (H.S.L.); leeverda@yuhs.ac (S.L.); 3Division of Infectious Diseases, Department of Internal Medicine, Gangnam Severance Hospital, Yonsei University College of Medicine, 211 Enjuro, Gangnam-gu, Seoul 06273, Korea; dayoung747@yuhs.ac (D.K.); subink93@yuhs.ac (S.K.); ayu870213@yuhs.ac (J.H.H.); shhan74@yuhs.ac (S.H.H.); imfell@yuhs.ac (Y.G.S.); 4Department of Pediatrics, Yonsei University College of Medicine, Seoul 03722, Korea; shinji@yuhs.ac

**Keywords:** pulmonary tuberculosis, lung cancer, meta regression, burden of tuberculosis

## Abstract

Pulmonary tuberculosis (TB) is a known risk factor for lung cancer. However, a detailed analysis of lung cancer type, age, sex, smoking, and TB burden associated with geographic and socioeconomic status has not been performed previously. We systematically appraised relevant observational studies reporting an association between pulmonary TB and lung cancer. All studies were included in the primary analysis, and studies that used robust TB diagnostic methods, such as validated medical diagnostic codes, were included in the secondary analysis. Thirty-two articles were included. The association between the history of pulmonary TB and diagnosis of lung cancer was statistically significant (OR 2.09, 95% CI: 1.62–2.69, *p* < 0.001). There was a high heterogeneity (*I*^2^ = 95%), without any publication bias. The analysis indicated a high association in advanced articles describing stringent pulmonary TB diagnosis (OR 2.26, 95% CI: 1.29–3.94, *p* = 0.004). The subgroup analyses suggested a significant association in countries with medium or high TB burdens, from East Asia and the Pacific region, and upper-middle income countries. Heterogeneity within the subgroups remained high in a majority of the subgroup analyses. A meta-regression analysis revealed that younger patients showed a significantly higher association between TB and lung cancer (regression coefficient = 0.949, *p* < 0.001). The history of pulmonary TB is an independent risk factor for lung cancer, especially in younger patients diagnosed with pulmonary TB. Clinicians should be aware of this association while treating young patients with a history of pulmonary TB.

## 1. Introduction

Lung cancer is one of the most common malignancies, with approximately 2.09 million new diagnoses worldwide in 2018. It also accounts for approximately 18.4% of the total cancer-related deaths, the highest of all cancer types [1]. The prognosis of lung cancer is relatively unfavorable compared to that of other malignancies, and as a prognosis largely depends on the stage of onset, thus, the early diagnosis of lung cancer is very important.

Cigarette smoking has been a major causal factor for lung cancer since 1912 [2,3,4]. Environmental factors such as air pollution, nutrition, occupational exposure, and a family history of previous cancer are also related to lung cancer. With the recent development of molecular diagnosis, research on genetic or inflammatory factors that contribute to lung carcinogenesis is being actively conducted [5,6].

Chronic inflammation resulting in pathological changes is a major risk factor in carcinogenesis. Inflammation is known to play a key role in carcinogenesis, such as infection with hepatitis B and C viruses in hepatocellular carcinoma, *Helicobacter pylori* in gastric cancer, and human papilloma virus in gynecological cancers [7]. Several meta-analyses have shown that previous inflammatory diseases in the lungs, such as pneumonia, chronic bronchitis, and pulmonary tuberculosis (TB), may increase the risk of lung cancer (relative risk ratio 1.36–1.90), independent of cigarette smoking [8,9]. According to forty-nine studies, pulmonary and extra-pulmonary TB infections increase the risk of 10 cancer types, including head and neck cancer, leukemia, lymphoma, gastrointestinal cancer, kidney cancer, bladder cancer, and lung cancer [10]. Thus, TB infection may influence the pathogenesis of lung cancer with or without cigarette smoking. To prevent the emergence of airborne transmittable TB and its progression to cancer, the control and prevention of TB is very important.

A detailed analysis of lung cancer types, patient age, sex, smoking status, and TB burden associated with geographic and socioeconomic statuses has not been performed in previous studies. Therefore, this study aimed to clarify the association between previous pulmonary TB infection and lung cancer by performing a comprehensive review of selected high-quality studies. We systematically reviewed the relationship between TB and lung cancer and also assessed various subgroups of the study population to identify the factors that affect this causal relationship.

## 2. Materials and Methods

### 2.1. Literature Search Strategy and Eligibility Criteria

This study was prospectively registered at PROSPERO (registration number: CRD42020211014). Two researchers (J.Y.K. and S.Y.H.) independently searched the PubMed, EMBASE, and Cochrane databases on 30 August 2020, using “tuberculosis” and “lung cancer” (complete search strategy provided in Appendix A) as keywords. Observational studies reporting an association between a history of pulmonary TB and lung cancer were included; reviews, clinical trials, and case reports were excluded. Studies lacking an estimation of the relative risk or data necessary to calculate risk or that used a limited population with a history of environmental or occupational exposure such as asbestos and silica were omitted. Non-English articles and studies published before 1990 were also excluded. When there were two or more articles using overlapping data sources, the article with the largest number of participants was selected. Eligible studies were extracted by first screening the title and abstract, followed by the full text. References from the relevant articles were also reviewed for selecting eligible studies. Disagreements were resolved by discussions between S.Y.H., J.Y.K., and K.H.L.

### 2.2. Diagnosis of TB and Lung Cancer

Studies, regardless of TB diagnostic methods, such as historical interviews and imaging, such as chest radiography and computed tomography, were included in the primary analysis. We recognized the weakness of diagnosing TB by radiographic images, history, or questionnaire and tried an additional analysis with high-quality studies based on microbiological diagnosis. Thus, we defined studies as high-quality if they used robust diagnostic methods based on medical records, including International Classification of Diseases (ICD) codes: ICD-8 codes 011 (pulmonary TB) and 012 (other respiratory TB); ICD-9 codes 010 (primary tuberculous infection), 011 (pulmonary TB), 012 (other respiratory TB), and 018 (miliary TB); and ICD-10 codes A15 (respiratory TB, bacteriologically and histologically confirmed), A16 (respiratory TB, not confirmed bacteriologically or histologically), and A19 (miliary TB). Regardless of the lung cancer type, studies that diagnosed lung cancer based on validated records such as hospital medical records or national registries were included.

### 2.3. Data Extraction

From the eligible studies, we extracted the name of the first author; publication year; country; baseline population characteristics (mean age, sex, smoking history, comorbidities with diabetes, hypertension, and chronic obstructive pulmonary diseases); number of total participants; number of TB cases; number of lung cancer cases; diagnostic method of TB; lung cancer type (small cell carcinoma, adenocarcinoma, large cell carcinoma, and squamous cell carcinoma); study design; effect size metrics or data necessary for calculations; maximally adjusted effect estimate of the association; and covariates used for adjustment. The Newcastle–Ottawa Scale (NOS) was applied to assess the risk of bias in the observational studies [11]. Countries were classified by region and economic income as per the World Bank classifications [12]. The incidence, prevalence, and mortality of TB are published annually by the World Health Organization (WHO) using information gathered through surveillance systems. The TB burden was stratified by the WHO definition, and high-burden countries were defined based on the TB burden data during 2016–2020 as 20 countries with the highest estimated numbers of TB incidence plus the top 10 countries with the highest estimated TB incidence. Intermediate and low-burden countries were defined as countries with a TB incidence of >40 and <40 cases per 100,000 persons, respectively, according to the WHO TB burden estimates for 2019 [13].

### 2.4. Statistical Analysis

We pooled adjusted estimates from all eligible studies for the primary analysis and pooled adjusted estimates of high-quality studies for the secondary analysis. The random effects model was used to obtain the summary estimates [14]. Heterogeneity between the included studies was estimated by *I*^2^ statistics [15].

Subgroup analyses were performed by country group (geographic region, economic status, and TB burden); covariates used for estimate adjustment (age; sex; smoking status; hypertension; diabetes; and history of respiratory diseases, such as pneumonia, chronic obstructive pulmonary diseases (COPD), and emphysema); cohort type (population-based or hospital-based); study design (prospective cohort, retrospective cohort, or case–control study); and TB diagnostic method (medical record, imaging, and self-report or physical examination). Meta-regression analyses were performed using covariates of baseline population characteristics (mean age, sex, and history of COPD and smoking) and lung cancer type (adenocarcinoma, squamous cell carcinoma, small-cell lung cancer, and large-cell lung cancer).

Publication bias was examined through visual inspection of the funnel plot and Egger’s test [16]. In order to prevent overestimation due to data duplication when using overlapping data sources, articles with a high number of participants that meet the research purpose and have high statistical power were selected and then analyzed. Additionally, a sensitivity analysis was performed while changing the data source, but there was no significant difference in the results. All statistical tests were two-sided. Statistical analyses were performed using software R version 4.0.3 and its “metafor” package [17,18].

## 3. Results

### 3.1. Characteristics of Literature

In the initial search, 10,229 potentially eligible articles were identified, of which 138 studies were selected for text screening. Finally, 32 articles [19,20,21,22,23,24,25,26,27,28,29,30,31,32,33,34,35,36,37,38,39,40,41,42,43,44,45,46,47,48,49,50] corresponding to 33 cohorts met the inclusion criteria (Table 1). The reasons for exclusion are shown in Figure 1. The included studies comprised a total of 982,797 participants, of whom 30,159 had pulmonary TB and 52,773 had lung cancer. The participating countries were from the East Asia and Pacific, Europe, and Central Asia and North America regions. All the countries had a high- or upper-middle income status, as low- and low-middle income countries were not included in these studies. Eight out of 32 articles were included in the high-quality TB diagnosis, followed by a secondary in-depth analysis.

The diagnosis of pulmonary TB is definitively established by the isolation of *M. tuberculosis* from a clinical specimen or tissue. An acid-fast bacilli smear or a polymerase chain reaction-based diagnosis are also used. However, radiographic image is just important supportive diagnostic tool. Based on the microbiological and pathological diagnoses, ICD code-related TB is then confirmed by the physician. Therefore, six out of the 32 enrolled articles presented an ICD code of tuberculosis for diagnosis, and two articles showed diagnosis by a medical expert based on the same criteria. These eight articles were included in the high-quality articles in this study. The remaining 24 articles were excluded from the final high-quality analysis, because they depended on interviews, memories of the participating patients, and medical images.

### 3.2. Pulmonary TB and Risk of Lung Cancer with All Eligible Studies

The overall association between a previous history of pulmonary TB and newly diagnosed lung cancer was statistically significant (odds ratio (OR): 2.09; 95% confidence interval (CI): 1.62–2.69, *p* < 0.001). There was high heterogeneity (*I*^2^ = 95%), no evidence of publication bias, the egger *p*-value was 0.447, and no visual asymmetry in the funnel plot (Figure 2A and Figure 3A). In the subgroup analysis by TB burden, the high-burden countries showed higher OR (2.57, 95% CI: 1.68–3.93, *p* < 0.001) than the medium-burden (OR: 2.48, 95% CI: 1.71–3.58, *p* < 0.001) and low-burden countries (OR: 1.77, 95% CI: 1.22–2.56, *p* = 0.003). Geographically, East Asia and the Pacific region showed a prominent risk (OR: 2.49, 95% CI: 1.83–3.39, *p* < 0.001) compared to the Europe and Central Asia (OR: 1.60, 95% CI: 0.80–3.22, *p* = 0.185) or North America (OR: 1.53, 95% CI: 1.11–2.12, *p* = 0.010) regions. The economic income statuses of the countries also reflected the characteristics of patients with TB, and the countries with upper-middle incomes (OR: 2.57, 95% CI: 1.68–3.93, *p* < 0.001) demonstrated a higher risk of lung cancer than high-income status countries (OR: 1.91, 95% CI: 1.41–2.59, *p* < 0.001). The association between pulmonary TB and newly developed lung cancer was statistically significant regardless of the adjustment for age, sex, smoking status, and cohort type or study design. The magnitude of association was similar regardless of whether pulmonary TB was diagnosed based on medical records (OR: 2.26, 95% CI: 1.29–3.94, *p* = 0.004), imaging (OR: 2.13, 95% CI: 1.16–3.92, *p* = 0.015), or self-report/physical examination (OR: 1.96, 95% CI: 1.56–2.47, *p* < 0.001). The heterogeneity within subgroups remained at a high level in a majority of the subgroup analyses (Table 2).

### 3.3. Pulmonary TB and Risk of Lung Cancer with High-Quality Studies

The analysis of eight high-quality studies showed a higher OR (2.26, 95% CI: 1.29–3.94, *p* = 0.004) than the analysis of all the studies. There was a high heterogeneity (*I*^2^ = 99%) with no publication bias, with Egger *p* = 0.621, and no visual asymmetry in the funnel plot (Figure 2B and Figure 3B). Of the eight articles, seven had cohorts from countries with a low TB burden, and only one had a cohort from a country with a medium TB burden. In the subgroup analysis with a TB burden, the medium-burden countries showed higher OR (4.18, 95% CI: 3.15–5.55, *p* < 0.001) than the low-burden countries (OR: 2.04, 95% CI: 1.12–3.73, *p* = 0.020). Geographically, the East Asia and the Pacific region showed a more prominent risk (OR: 2.79, 95% CI: 1.21–6.39, *p* = 0.016) compared to the Europe and Central Asia regions (OR: 1.79, 95% CI: 0.67–4.77, *p* = 0.244) (Table 3).

### 3.4. Stratified and Sensitivity Analysis

The quality of the 33 included articles was evaluated using the NOS. The quality assessment of 27 case–control studies is shown in Table 4 and that of six retrospective cohort studies is demonstrated in Table 5. We performed meta-regression analyses with continuous variables, such as the mean age at diagnosis of pulmonary TB, baseline characteristics including comorbidity, and pathological cell type of lung cancer. All the results are shown in Appendix A. Of these, patients with a low mean age at diagnosis of pulmonary TB showed a significant association between pulmonary TB and lung cancer. The primary analysis with all 32 articles estimated a regression coefficient of 0.949 (*p* < 0.001). The secondary analysis with eight high-quality studies with stringent TB diagnostic methods showed similar results (regression coefficient = 0.945, *p* < 0.001) (Figure 4).

## 4. Discussion

This study assessed the association between previous pulmonary TB infection and the risk of lung cancer. The overall effect size of the eight high-quality studies was higher than that of the 32 total studies. The risk increased in patients diagnosed with pulmonary TB at a young age and in countries with a high TB burden, upper-middle income economic status, and the East Asian and Pacific regions.

Reverse causality between pulmonary TB and lung cancer is an unaddressed issue. To establish a temporal order from TB diagnosis to lung cancer diagnosis, we included case–control studies of cases with lung cancer and cohort studies following TB and non-TB populations. In addition, diagnostic misclassification between pulmonary TB and lung cancer occurs because of overlapping radiologic findings and the concomitant existence of the two entities [51]. Moreover, the similarity in clinical symptoms, such as cough, expectoration, fever, hemoptysis, and weight loss, as well as radiological similarities, contribute to misdiagnosis of the two diseases [52]. This has frequently led to the misdiagnosis of lung cancer as TB and prescribed medication for TB in patients with lung cancer in some countries [53]. Since extra-pulmonary TB is often misdiagnosed as cancer in its early stages, the difficulty of diagnosis is also applied in the opposite case [54]. To account for the potential bias due to the diagnostic misclassification of pulmonary TB with other respiratory diseases, we performed subgroup analyses of eight high-quality studies that diagnosed TB with validated medical codes; the analyses suggested that the observed association was similar to that in the main analysis.

The increased risk of lung cancer found in younger patients with TB was described by Wu et al. and An et al., both included in previous meta-analyses [20,55]. However, an overall unfavorable opinion on the association between age and increased risk warrants additional analysis.

The increased risk of pulmonary TB in East Asia and Pacific countries may be associated with socioeconomic determinants, such as hunger and limited public health infrastructure, that are related to poor TB outcomes [56,57]. The control of TB is poorer in these regions, with as a higher TB burden, or upper-middle income countries due to the difficulty in accessing healthcare. The burden of TB and the national income tend to be inversely proportional. Since these countries lack a necessary medical infrastructure, the early diagnosis of TB is difficult, the incidence of multidrug-resistant TB is high, and latent TB infection (LTBI) cannot be managed. A study in Indonesia suggested that the diagnosis of pulmonary TB was delayed in a majority of patients by >30 days after the onset of symptoms and that those patients were also unaware of the disease [58]. In 2019, the largest proportion of newly diagnosed TB cases (44%) was observed in the South-East Asia region, and the attributable risk factors were undernutrition, infection with human immunodeficiency virus, alcoholism, smoking, and diabetes [59].

Smoking is the most important environmental risk factor for lung cancer and is also known to increase the risk of TB [60]. However, in the present study, the current smoking status failed to correlate with an increase in the risk of lung cancer, confirming that smoking was not a confounding factor in this progression. The findings of the present study are consistent with those of a previous systematic review by Liang et al., which suggested that smoking is not an influential factor in the increased risk of lung cancer in patients with preexisting TB [61]. Moreover, there was no significant relationship with a history of common comorbidities, such as hypertension and diabetes.

There was no significant association between the incidence of TB and occurrence of a particular lung cancer type. This is in contrast to the results of a meta-analysis of 41 studies that showed a significant association between TB infection and adenocarcinoma [61]. Another study provided experimental evidence that chronic TB leads to the remodeling of lung tissue, evoking squamous cell metaplasia and a microenvironment accelerating squamous cell carcinogenesis [62]. However, the present study showed there was no meaningful association between TB and lung cancer type.

Several causal mechanisms have been suggested to explain this association. First, chronic inflammation induced by pulmonary TB may induce genetic mutations in lung parenchyma cells [53]; this hypothesis is supported by evidence that chronic TB induces cell dysplasia and the aggregation of squamous cells through *Mycobacterium tuberculosis*-infected macrophages [62]. This also applies to LTBI, which consists of chronic infection and inflammation, thus leading to an increased risk of lung cancer. The patients with LTBI showed a higher risk of future lung cancer (hazard ratio (HR) 2.69, 95% CI 0.66–11.07, *p* = 0.170). Additionally, among 136 TB contacts who received isoniazid prophylaxis, not one developed cancer. Therefore, the treatment of TB, including LTBI, is important in the further prevention of cancer by the suppression of chronic inflammation [63]. Second, the immunosuppressive role of TB by naturally occurring regulatory T cells may increase the risk of malignancy [64]. Finally, *Mycobacterium tuberculosis* can induce cellular DNA damage by increasing exposure to reactive oxygen and reactive nitrogen intermediates while entering macrophages [65]. Moreover, patients with TB showed frequent DNA damage, defects in cytokinesis, and a higher frequency of apoptotic and necrotic cells [66]. Additionally, repeated tissue damage cause fibrotic scar tissue formation; then, fibrosis could promote an enhanced tumorigenic potential [67].

The link between chronic inflammation in pulmonary TB and tumorigenesis of lung cancer possesses further findings applicable in clinical settings. Potential biomarkers such as nuclear protein *MKI67,* which regulates the genes related to tumorigenesis, can predict lung cancer, as it is shared by both TB and lung adenocarcinoma [68]. TB-induced EGFR mutations and epiregulin, a potent ligand for EGFR, may also contribute to carcinogenesis, and the treatment response of EGFR-TKIs differs in patients with pulmonary adenocarcinoma who previously had TB [62,69].

The present study had several limitations. First, this was a retrospective study that depended on ICD codes. Previous experimental studies have identified an association between pulmonary TB and progression to squamous cell lung cancer; however, the present retrospective study failed to provide any evidence on the correlation of a particular lung cancer type with TB. Thus, well-designed prospective studies are warranted to validate the association. Second, we could not identify individual studies from low-middle income or low-income countries, and in the analysis of the high-quality studies, the countries with a low TB burden accounted for seven out of eight studies. The absence of data from countries with low incomes and high TB burdens may have introduced bias in the summary estimate and lead to discrepancies in real-world estimates. Lastly, an analysis of treatment outcomes such as resistant TB, drug adherence, and treatment failure was not included. As this study reviewed articles analyzing the relationship between the risk factors of pulmonary tuberculosis and lung cancer, the focus was mainly on the “diagnosis”. In general, the incidence of resistant tuberculosis is high in countries with a low socioeconomic status, and in these areas, treatment failure and drug adherence are also poor. Therefore, this will also be a good additional analytic point in the future.

Nevertheless, the study had some strengths. Several studies have investigated the association between TB infection and lung cancer. However, the present study compared the results of all the studies and a few high-quality studies and obtained consistent results. Further, while screening, we excluded patient groups with specific diseases to maintain homogeneity in the complete population. We also merged the data of various subgroups to identify the characteristics that affected this association, and we obtained several significant results. Based on the socioeconomic factors and TB infection, which are inseparable, we elucidated that the patients in countries with high s TB burden, upper and middle incomes, and from the East Asia and Pacific regions had a higher risk of lung cancer than other patients. We believe that these findings will help design an approach for prioritizing and planning a strategy for worldwide TB control [70].

## 5. Conclusions

In conclusion, we found that the diagnosis of pulmonary TB at a young age is a risk factor for lung cancer, regardless of the underlying disease or smoking history. This trend is more pronounced in patients from countries with a high TB burden. This study is significant, as it summarized the results of several observational studies that described the association between pulmonary TB and lung cancer.

## Figures and Tables

**Figure 1 jcm-11-00765-f001:**
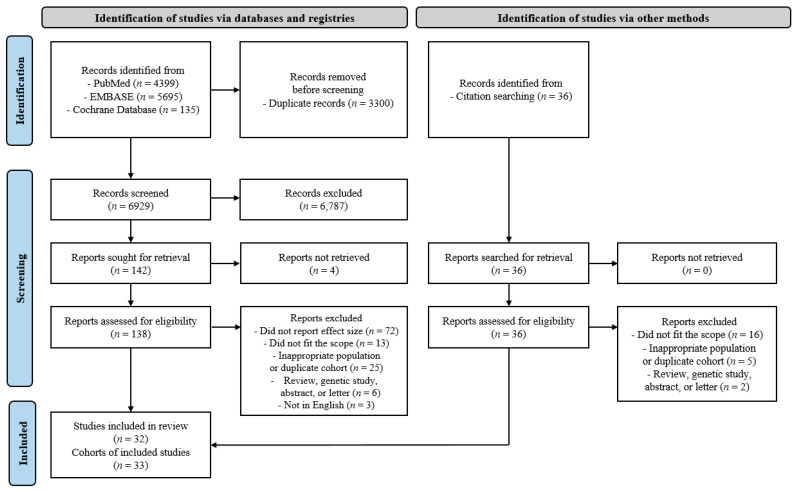
Flow chart of the included studies.

**Figure 2 jcm-11-00765-f002:**
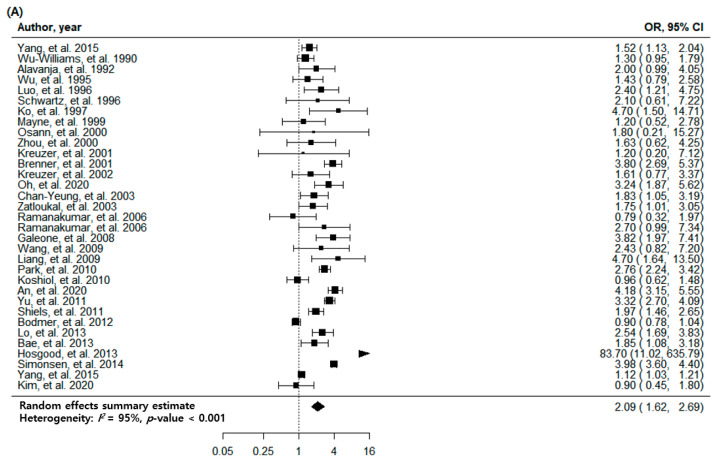
Forest plots of risk estimates for the association between tuberculosis and lung cancer. (**A**) Meta-analysis of all eligible studies. (**B**) Meta-analysis of high-quality studies.

**Figure 3 jcm-11-00765-f003:**
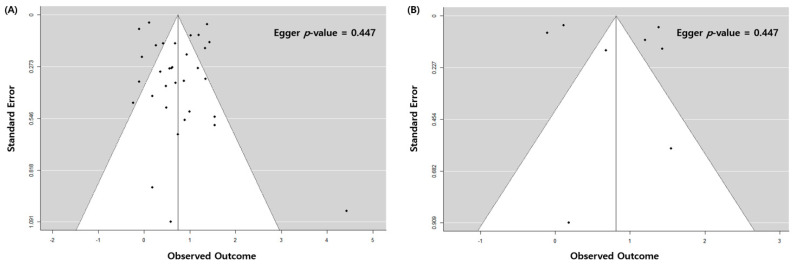
Funnel plot of the study estimates. (**A**) All eligible studies. (**B**) High-quality studies.

**Figure 4 jcm-11-00765-f004:**
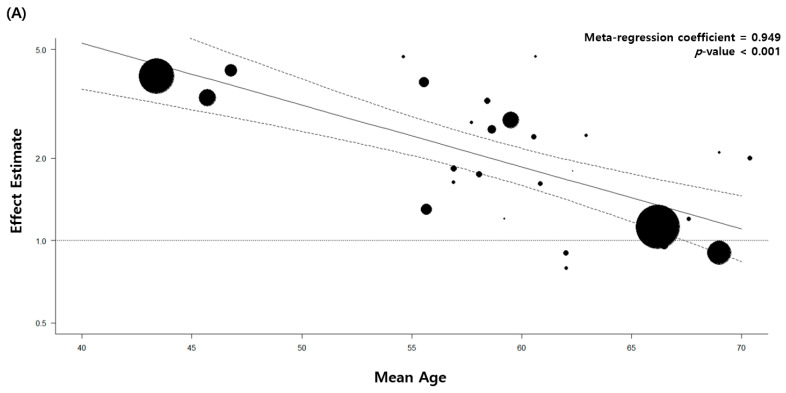
Meta-regression analysis of the mean patient age and association between tuberculosis and lung cancer. (**A**) All eligible studies. (**B**) High-quality studies.

**Table 1 jcm-11-00765-t001:** Baseline characteristics of the studies included in the meta-analysis.

First Author, Year	Country	No. of Participants	No. of pul. TB	No. of Lung Cancer	Characteristics of Patients	Diagnostic Method of pul. TB	Study Design	Cohort	Adjusted Covariates
Kim et al. 2020 [19]	South Korea	11,394	1509	65	Smokers	Low dose CT	Prospective cohort	Population-based	Age, sex, smoking status, smoking burden, family history of lung cancer, nodule count per CT scan, and comorbidities (pulmonary fibrosis and emphysema)
An et al. 2020 [20]	South Korea	22,656	3776	194	General population	ICD-10 codes (A15, A16, A19)	Case-control	Population-based	Age, sex, smoking status, and income level
Oh et al. 2020 [21]	South Korea	20,252	2640	65	General population, aged ≥40 years	Medical history and chest X-ray	Retrospective cohort	Population-based	Age, sex, education, income level, smoking status, BMI, and physical activity
Yang et al. 2015 [22]	China	3238	199	1559	General population	Interview	Case-control	Hospital-based	Age, sex, BMI, education, smoking status, smoking burden, exposure to passive smoking, occupational exposure to metallic toxicant, housing ventilation, intakes of cured meat and vegetables or fruit, and emphysema
Yang et al. 2015 [23]	Taiwan	34,658	1455	17,329	Women	ICD-9 codes (010–012, 018)	Case-control	Population-based	Age, income level, employment, and comorbidities (asthma, COPD, hypertension, and stroke)
Simonsen et al. 2014 [24]	Denmark	11,472	11,472	390	General population	ICD-8 codes (011, 012) and ICD-10 codes (A15, A16)	Retrospective cohort	Population-based	Age, sex, comorbidities, and patients’ country of origin
Hosgood et al. 2013 [25]	China	996	26	498	General population	Interview	Case-control	Population-based	Sex, education, family history of lung cancer, occupation, smoking status, exposure to passive smoking, and exposure to household heating and cooking fume
Bae et al. 2013 [26]	South Korea	7009	658	93	Men	Questionnaire	Prospective cohort	Population-based	Age and intake of coffee and tomato
Lo et al. 2013 [27]	Taiwan	3080	127	1540	Never smokers	Interview	Case-control	Hospital-based	Age and education
Bodmer et al. 2012 [28]	United Kingdom	91,301	1621	13,043	Diabetes patients	UK-based General Practice Research Database (GPRD)	Case-control	Population-based	Age, sex, BMI, smoking, and comorbidities (diabetes and COPD)
Shiels et al. 2011 [29]	Finland	29,133	273	3102	General population	ICD-9 codes (010-012)	Prospective cohort	Population-based	Age and smoking status
Yu et al. 2011 [30]	Taiwan	716,872	4480	1684	General population	ICD-9 (011)	Prospective cohort	Population-based	Age, sex, occupation, and comorbidities (hypertension, dyslipidemia, diabetes, and COPD)
Koshiol et al. 2010 [31]	Italy	3968	121	1890	General population	Interview	Case-control	Population-based	Age, sex, region, smoking duration, and chronic bronchitis
Park et al. 2010 [32]	South Korea	4916	759	3781	General population	Questionnaire and chest X-ray	Case-control	Hospital-based	Age and smoking status
Liang et al. 2009 [33]	China	505	21	226	Never smoker, women	Interview	Case-control	Hospital-based	Age, marital status, education, ethnicity, BMI, coal use, exposure to passive smoking, and exposure to coal smoke and cooking fumes
Wang et al. 2009 [34]	Hong Kong	504	18	212	Females, aged 30–79 years	Interview	Case-control	Hospital-based	Age, occupation, exposure to cooking, and intakes of vegetables and vitamins
Galeone et al. 2008 [35]	China	654	47	216	General population	Interview	Case-control	Hospital-based	Age, sex, region, smoking status, smoking burden, income, family history of lung and other cancers, and occupational exposure to lung carcinogen
Ramanakumar et al. 2006 [36]^ a^	Canada	2461	68	755	General population	Interview	Case-control	Population-based	Age, ethnicity, type of respondent, education, income level, and smoking status
Ramanakumar et al. 2006 [36]^ b^	Canada	2746	56	1205	General population	Interview	Case-control	Population-based	Age, ethnicity, type of respondent, education, income level, and smoking status
Zatloukal, et al. 2003 [37]	Czech Republic	1990	128	366	Women	Questionnaire	Case-control	Hospital-based	Age, region, education, and smoking burden
Chan-Yeung et al. 2003 [38]	China	662	72	331	General population	Questionnaire	Case-control	Hospital-based	Smoking burden
Kreuzer et al. 2002 [39]	Germany	769	31	234	General population	Interview	Case-control	Population-based	Age and region
Brenner et al. 2001 [40]	China	2651	162	886	General population	Interview	Case-control	Population-based	Age, sex, region, and smoking status
Kreuzer et al. 2001 [41]	Germany	861	38	58	General population	Confrimed by physician	Case-control	Population-based	Age and region
Zhou et al. 2000 [42]	China	144	25	72	Women	Interview	Case-control	Population-based	Age, marital status, education, and BMI
Osann et al. 2000 [43]	U.S.A.	302	8	98	Women	Interview	Case-control	Hospital-based	Age, education, and smoking
Mayne et al. 1999 [44]	U.S.A.	874	22	437	Non-smokers	Interview	Case-control	Population-based	Smoking, exposure to passive smoking, and comorbidities (emphysema, chronic bronchitis, and asthma)
Ko et al. 1997 [45]	Taiwan	210	20	105	General population	Confrimed by physician	Case-control	Hospital-based	Socioeconomic status, region, and education
Schwartz et al. 1996 [46]	U.S.A.	534	12	257	Non-smoker, African Americans and Caucasians	Interview	Case-control	Population-based	Age, sex, and ethnicity
Luo et al. 1996 [47]	China	408	39	102	General population	Interview	Case-control	Population-based	Age, sex, and ethnicity
Wu et al. 1995 [48]	U.S.A.	1633	56	397	Non-smoker, women	Interview	Case-control	Population-based	Age, ethnicity, region, education, comorbidities (lung diseases including asthma, chronic bronchitis, pneumonia, pleurisy, and emphysema)
Alavanja et al. 1992 [49]	U.S.A.	2020	34	618	White, non-smoking, women	Interview	Case-control	Population-based	Age and smoking
Wu-Williams et al. 1990 [50]	China	1924	186	965	General population	Interview	Case-control	Population-based	Age, education, study area, and smoking status

^a, b^: Two separate cohorts reported in one article. Study ^a^ was conducted in 1979–1986 (755 cases and 512 controls); study ^b^ was conducted in 1996–2001 (1205 cases and 1541 controls). Abbreviations: BMI, body mass index; COPD, chronic obstructive pulmonary disease; CT, computed tomography; ICD, International Classification of Diseases; No., number; pul.TB, pulmonary tuberculosis; U.S.A., United States of America.

**Table 2 jcm-11-00765-t002:** Meta-analysis of 33 eligible cohorts to assess the association between pulmonary tuberculosis and lung cancer.

Subgroup	No. of Cohorts *	OR (95% CI)	*p*-Value	*I^2^* Value (%)	*I*^2^ between Subgroups (%)
**All cohorts**	33	2.09 (1.62–2.69)	<0.001	95	
**TB burden of country**					
Low	18	1.77 (1.22–2.56)	0.003	97	12
Medium	6	2.48 (1.71–3.58)	<0.001	75	
High	9	2.57 (1.68–3.93)	<0.001	81	
**Region of country**					
East Asia and Pacific	19	2.49 (1.83–3.39)	<0.001	93	58
Europe and Central Asia	7	1.60 (0.80–3.22)	0.185	98	
North America	7	1.53 (1.11–2.12)	0.010	0	
**Economic status of country**					
High-income	24	1.91 (1.41–2.59)	<0.001	96	20
Upper-middle-income	9	2.57 (1.68–3.93)	<0.001	81	
**Age**					
Adjusted	29	2.00 (1.54–2.61)	<0.001	95	14
Not adjusted	4	3.84 (1.21–12.15)	0.022	82	
**Sex**					
Adjusted	22	2.23 (1.60–3.11)	<0.001	96	0
Not adjusted	11	1.90 (1.47–2.46)	<0.001	61	
**Smoking**					
Adjusted	22	2.03 (1.51–2.73)	<0.001	90	0
Not adjusted	11	2.19 (1.34–3.59)	0.002	98	
**Hypertension**					
Adjusted	2	1.92 (0.66–5.57)	0.230	99	0
Not adjusted	31	2.10 (1.62–2.73)	<0.001	92	
**Diabetes**					
Adjusted	2	1.72 (0.48–6.20)	0.404	99	0
Not adjusted	31	2.13 (1.63–2.77)	<0.001	94	
**Respiratory comorbidities**					
Adjusted	8	1.32 (0.93–1.86)	0.121	94	90
Not adjusted	25	2.51 (2.04–3.08)	<0.001	78	
**Cohort of the study**					
Population-based	23	1.95 (1.41–2.68)	<0.001	96	0
Hospital-based	10	2.36 (1.85–3.01)	<0.001	49	
**Study design**					
Prospective cohort study	4	1.96 (1.22–3.15)	0.005	84	94
Retrospective cohort study	2	3.95 (3.58–4.36)	<0.001	0	
Case-control study	27	1.99 (1.56–2.53)	<0.001	89	
**Diagnostic method** **of pulmonary TB**					
Medical record	8	2.26 (1.29–3.94)	0.004	99	0
Imaging	3	2.13 (1.16–3.92)	0.015	80	
Self-report or physical examination	22	1.96 (1.56–2.47)	<0.001	66	

* Since two separate cohorts were reported in one article, a total of 33 eligible cohorts were extracted and analyzed from 32 enrolled studies. Abbreviations: CI, confidence interval; No, Number; OR, odds ratio; TB, tuberculosis.

**Table 3 jcm-11-00765-t003:** Meta-analysis of high-quality studies to assess the association between TB and lung cancer.

Subgroup	No. of Studies	OR (95% CI)	*p*-Value	*I^2^* Value (%)	*I*^2^ between Subgroups (%)
**All studies**	8	2.26 (1.29–3.94)	0.004	99	
**Country of TB burden**					
Low	7	2.04 (1.12–3.73)	0.020	99	78
Medium	1	4.18 (3.15–5.55)	<0.001	-	
High	0	-	-	-	
**Region of country**					
East Asia and Pacific	4	2.79 (1.21–6.39)	0.016	98	0
Europe and Central Asia	4	1.79 (0.67–4.77)	0.244	99	
North America	0	-	-	-	

Abbreviations: CI, confidence interval; No, Number; OR, odds ratio; TB, tuberculosis.

**Table 4 jcm-11-00765-t004:** Quality assessment of the included case–control studies using the Newcastle–Ottawa Scale.

Study	Selection				Comparability	Outcome			Quality Score
	Adequacy of Case Definition	Degree of Representation of Cases	Selection of Controls	Definition of Controls	Comparability of Cases and Controls on the Basis of Design or Analysis	Confirmation of Exposure	Same Method of Confirmation for Cases and Controls	Non-Response Rate	
An et al. 2020 [20]	*	*	*	*	**	*	*	*	9
Yang et al. 2015 [22]	*	*	*	*	**		*	*	8
Yang et al. 2015 [23]	*	*	*	*	*	*	*	*	8
Hosgood et al. 2013 [25]	*	*	*	*	*		*	*	7
Lo et al. 2013 [27]	*	*	*	*	**		*	*	8
Bodmer et al. 2012 [28]	*	*	*	*	**	*	*	*	9
Koshiol et al. 2010 [31]	*	*	*	*	**		*	*	8
Park et al. 2010 [32]	*	*	*	*	**		*		7
Liang et al. 2009 [33]	*	*	*	*	**		*	*	8
Wang et al. 2009 [34]	*	*	*	*	*		*	*	7
Galeone et al. 2008 [35]	*	*	*	*	**		*	*	8
Ramanakumar et al. 2006 [36] ^a^	*	*	*	*	**		*	*	8
Ramanakumar et al. 2006 [36] ^b^	*	*	*	*	**		*	*	8
Zatloukal et al. 2003 [37]	*	*	*	*	**		*	*	8
Chan-Yeung et al. 2003 [38]	*	*	*	*	*		*	*	7
Kreuzer et al. 2002 [39]	*	*	*	*	*		*	*	7
Brenner et al. 2001 [40]	*	*	*	*	**		*	*	8
Kreuzer et al. 2001 [41]	*	*	*	*	*	*	*	*	8
Zhou et al. 2000 [42]	*	*	*	*	*		*	*	7
Osann et al. 2000 [43]	*	*	*	*	**		*	*	8
Mayne et al. 1999 [44]	*	*	*	*	*		*	*	7
Ko et al. 1997 [45]	*	*	*	*		*	*	*	7
Schwartz et al. 1996 [46]	*	*	*	*	**		*	*	8
Luo et al. 1996 [47]	*	*	*	*	*		*	*	7
Wu et al. 1995 [48]	*	*	*	*	**		*	*	8
Alavanja et al. 1992 [49]	*	*	*	*	**		*	*	8
Wu-Williams et al. 1990 [50]	*	*	*	*	**		*	*	8

^a,b^: Two separate cohorts reported in one article. Study ^a^ was conducted in 1979–1986 (755 cases and 512 controls); study ^b^ was conducted in 1996–2001 (1205 cases and 1541 controls). A study can be awarded a maximum of one star for each numbered item within the Selection and Outcome categories. A maximum of two stars can be given for Comparability.

**Table 5 jcm-11-00765-t005:** Quality assessment of the included retrospective cohort studies using the Newcastle–Ottawa Scale.

	Selection				Comparability	Outcome			Quality Score
Study	Degree of Representation of the Exposed Cohort	Selection of the Non-Exposed Cohort	Confirmation of Exposure	Demonstration That the Current Outcome of Interest Is Absent at the Start of the Study	Comparability of Cohorts Based on Design or Analysis	Assessment of Outcome	Sufficiency of Follow-Up to Detect Outcomes	Adequacy of Follow-Up of Cohorts	
Kim et al. 2020 [19]	*	*		*	**	*	*		7
Oh et al. 2020 [21]	*	*		*	**	*	*	*	8
Simonsen et al. 2014 [24]	*	*	*	*	*	*	*	*	8
Bae et al. 2013 [26]	*	*		*	**	*	*		7
Shiels et al. 2011 [29]	*	*	*	*	**	*	*	*	9
Yu et al. 2011 [30]	*	*	*	*	*	*	*		7

A study can be awarded a maximum of one star for each numbered item within the Selection and Outcome categories. A maximum of two stars can be given for Comparability.

## Data Availability

Not applicable.

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
