# Peer review of "Pulmonary Tuberculosis and Risk of Lung Cancer: A Systematic Review and Meta-Analysis"

_jcm, 2022, doi:10.3390/jcm11030765_

Round 1

Reviewer 1 Report

In this manuscript, Hwang et al. have performed a meta-analysis of tuberculosis and lung cancer. This is an interesting study to point out the significant risk factor due to pulmonary tuberculosis. It is important because developing countries have a higher burden of tuberculosis and cancer. The manuscript structure is good but can be improved. Further, minor corrections in English and sentence structuring would be helpful. Here are my comments:

In methodology "When there were two or more articles using overlapping data sources, the article with the largest number of participants was selected." Sometimes larger studies exclude some patients from the smaller studies based on their own inclusion criteria. Did the authors carefully assess this variation?

  1. Authors can look into the addition of more epidemiological information in the introduction section. For example, the numbers on tuberculosis and extrapulmonary tuberculosis can be added.
  2. The author can improve this sentence for clarity. "However, in recent decades, as the characteristics of patients with lung cancer are changing, studies are focusing on various causes of lung cancer."
  3. In the following sentence, the authors can use 'smoking'. "

Cigarette remains a major risk factor, but environmental factors, such as air pollution, nutrition, family history of previous cancer, and genetic factors also contribute to lung carcinogenesis [5,6]."

  1. The following sentence can be restructured " Additionally, inflammation, characterized by chronic inflammation and pathological changes, is a major risk factor in carcinogenesis."
  2. In the flowchart "Figure 1. Flow chart of the included studies." the authors can correct the error in Embase.
  3. "Several causal mechanisms have been suggested to explain this association. First, chronic inflammation induced by pulmonary TB may induce genetic mutations in lung parenchyma cells" Authors can expand this sentence with additional details.
  4. The following sentence can be improved especially with respect to apoptotic cells. "Moreover, patients with TB showed frequent DNA damage, defect in cytokinesis, and apoptotic cells"

  1. For a better perspective author can provide information associated with extra-pulmonary tuberculosis and cancer. Extra-pulmonary tuberculosis is difficult to detect and also causes misdiagnosis. For example, following papeR says "Extrapulmonary tuberculosis was relatively common in younger patients with active M. tuberculosis infection, and was often initially misdiagnosed as cancer." rEFER:  Aisenberg, G.M., Jacobson, K., Chemaly, R.F., Rolston, K.V., Raad, I.I. and Safdar, A., 2005. Extrapulmonary tuberculosis active infection misdiagnosed as cancer: Mycobacterium tuberculosis disease in patients at a Comprehensive Cancer Center (2001–2005). Cancer: Interdisciplinary International Journal of the American Cancer Society104(12), pp.2882-2887.
  2. The authors can add a discussion on the need for prognostic/predictive biomarkers. These biomarkers can be nucleic acids, proteins, or other biological entities that can classify disease subgroups or at the intended end point. The authors of this journal can benefit from emerging molecular players at the intersection of chronic infection and cancer. For example in the following study: the authors have identified a transcriptional gene signature that can classify tuberculosis, lung adenocarcinoma, and sarcoidosis patients. They identified the deposits of extra-cellular matrix, and fibrillar collagen to be of similar composition in TB, lung cancer, and sarcoidosis.
  3. Most interestingly, this study identified overexpression of Marker Of Proliferation Ki-67 (MKI67), a tumor-promoting nuclear protein in both tuberculosis and lung cancer. Refer: Chai, Qiyao, Zhe Lu, Zhidong Liu, Yanzhao Zhong, Fuzhen Zhang, Changgen Qiu, Bingxi Li et al. "Lung gene expression signatures suggest pathogenic links and molecular markers for pulmonary tuberculosis, adenocarcinoma, and sarcoidosis." Communications biology 3, no. 1 (2020): 1-17.
  4. Another point that can enrich the discussion is the factor of latent tuberculosis and cancer. They identified a higher risk of Cancer in LTBI patients with TB activation (Hazard Ratio of 2.29) than in the control population. The author can find more information here: Su, V.Y.F., Yen, Y.F., Pan, S.W., Chuang, P.H., Feng, J.Y., Chou, K.T., Chen, Y.M., Chen, T.J. and Su, W.J., 2016. Latent tuberculosis infection and the risk of subsequent cancer. Medicine95(4).

Reviewer 2 Report

I want to congratulate the authors for the work done. This review and meta-analysis establishes a statistical relationship between the history of pulmonary tuberculosis and the presence of lung cancer, and also attempts to establish a causal relationship by exposing the possible mechanism that leads from one condition to another and the temporal order. The introduction summarizes well the previous knowledge on this field; the methodology is rigorous and well described; the results are clearly represented, and the discussion is extensive.

In my opinion, this review article merits publication. I only have a few minor comments:

1. The authors describe that one of the limitations is the retrospective nature of this study and the use on ICD codes. They also defend that in some cases there may be misclassification of diseases. Table 1 shows the TB diagnostic method of each study, where several studies have used ICD coding or radiological diagnosis. The diagnosis of pulmonary tuberculosis is a microbiological diagnosis, with a microbiological culture or a PCR, which not only serves to confirm the diagnosis, but also to guide treatment based on drug sensitivity. Therefore, a diagnosis based simply on radiological data does not guarantee a certain diagnosis. In the same way, the ICD codes also do not give a certain diagnosis if no microbiological method has been used. Therefore, I wanted to know how many studies, and especially the high-quality studies, have selected participants based on a microbiological diagnosis.

2. Do you have data on the treatment of pulmonary tuberculosis in the patients included in these studies? The correct pharmacological treatment and therapeutic compliance is also a factor that should be considered in the management of this disease, and it could vary from one country to another depending on the geographical area and the socioeconomic level. Following the same mechanism explained in the Discussion section, is it possible that patients who did not receive correct treatment or did not comply with treatment have a higher risk of developing lung cancer?

3. A total of 32 studies and 33 cohorts were included. However, Table 2 shows that there are 33 studies. Please correct this mistake.

Round 2

Reviewer 1 Report

The authors have improved the manuscript. I feel that authors should expand the segment on latent tuberculosis as this forms the vast 'reservoir' of potential patients.  
